# Reducing Human Effort in Ontology-Based Data Integration: An LLM-Assisted Mapping Pipeline

Sulthoni Ashiddiiqi[1,*,†], Fajar J. Ekaputra[2,†], Muhammad Zuhri Catur Candra[1,†] and Gusti Ayu Putri Saptawati Soekidjo[1,†]

[1]*School of Electrical Engineering and Informatics, Institut Teknologi Bandung (SEEI-ITB), Bandung, Indonesia*
[2]*Institute of Data, Process, and Knowledge Management (DPKM), WU Wien, Vienna, Austria*

## Abstract

Ontology-Based Data Integration (OBDI) relies on semantic mappings between data sources and a target ontology to provide uniform access to heterogeneous data. However, creating these mappings is complex, time-consuming, and resource-intensive, often requiring significant human expertise. This paper introduces LAMP (LLM-Assisted Mapping Pipeline), a novel approach that leverages Large Language Models (LLMs) to assist the mapping process in OBDI. LAMP decomposes the mapping task into manageable subtasks, leveraging retrieval-augmented techniques, contextual sufficiency management, and coherence maintenance across subtasks to enhance LLM performance. The pipeline incorporates a human-in-the-loop component for quality assurance. We evaluate LAMP on the BLINKG dataset across its three mapping scenarios, demonstrating significant improvements in F1-scores–particularly in complex mapping scenario–compared to single-prompt approach. Our findings highlight LAMP's potential to reduce human effort and improve efficiency in OBDI mapping tasks.

## Keywords

Ontology-Based Data Integration, mapping, Large Language Models.

## 1. Introduction

Ontology-Based Data Integration (OBDI) is a data integration approach that employs an ontology as the mediated schema. This paradigm offers several advantages: it enforces standardized terminology and conceptual structures, provides richer semantic representation through axioms and explicit relationships between entities, and enables automated inference to derive additional knowledge from the integrated data [1, 2]. Despite its many strengths, the practical implementation of OBDI—particularly in the development of the ontology and the construction of mappings between data sources and the ontology—remains complex, time-consuming, and resource-intensive [3, 4, 5, 6]. Both processes rely heavily on domain experts and ontology engineers, and are frequently executed manually. As the number and diversity of data sources increase, the mapping process becomes progressively more challenging, leading to higher costs, longer development cycles, greater risks of miscommunication, and significant scalability concerns for OBDI deployments [5, 6]. These factors collectively create a substantial barrier to adoption and hinder the broader practical implementation of OBDI [6].

The rapid advancement of Large Language Models (LLMs) in recent years has created a promising opportunity to reduce, and in some cases substitute, the reliance on domain experts and ontology engineers. Having been trained on vast and diverse corpora, LLMs possess broad cross-domain knowledge. Their growing capabilities in reasoning, contextual understanding, disambiguation of terminology, relationship inference, semantic interpretation, and semantic type detection offer a strong foundation for supporting mapping [5, 7]. Nevertheless, leveraging LLMs for OBDI mapping still presents several

*KGCW'26: 7th International Workshop on Knowledge Graph Construction, May 10th, 2026, Dubrovnik, Croatia*

*Corresponding author.

†These authors contributed equally.

✉ 33221006@std.stei.itb.ac.id (S. Ashiddiiqi); fajar.ekaputra@wu.ac.at (F. J. Ekaputra); catur@staff.stei.itb.ac.id (M. Z. C. Candra); putri@staff.stei.itb.ac.id (G. A. P. S. Soekidjo)

🌐 https://juang.id (F. J. Ekaputra)

🆔 0009-0009-9700-9266 (S. Ashiddiiqi); 0000-0003-4569-2496 (F. J. Ekaputra); 0000-0001-8260-2627 (M. Z. C. Candra); 0009-0006-5003-9781 (G. A. P. S. Soekidjo)

challenges, including token limitations, the risk of hallucination, and the need to design appropriate task decomposition strategies—tasks that are neither too large to exceed model capacity nor too small to lose coherence [5, 7, 8].

This study presents **LAMP** (an LLM-Assisted Mapping Pipeline). LAMP is designed to support the semantic mapping process in OBDI by leveraging LLM capabilities while remaining aware of their strengths and inherent limitations. The contributions of this paper are as follows:

1. A novel task-decomposed, LLM-based pipeline for semantic mapping in OBDI.
2. A hybrid methodology that integrates ontology extraction, vector retrieval, and LLM-driven reasoning to generate semantic mappings.
3. A reusable system architecture built upon LangChain, ChromaDB, and structured prompting techniques.

The structure of this paper is as follows: Chapter 2 reviews the related work. Chapter 3 presents the methodology, including the motivation for the mapping design and an overview of the proposed pipeline. Chapter 4 reports the evaluation of LAMP using experiments on the BLINKG dataset. Chapter 5 discusses the key findings, limitations, and implications. Finally, Chapter 6 provides the conclusion and outlines directions for future work. We provided a comparison of these existing studies with LAMP, along with the identified research gaps in Table 1.

## 2. Related Work

Research on mapping processes has advanced from multiple perspectives. From the declarative side (mapping languages), a wide range of RDF- and SPARQL-based languages have been developed, including D2RQ[11], R2RML[12], RML[13], and YARRRML [14]. Tooling innovations have generally advanced in parallel with these languages and techniques, as reflected in the emergence of mapping engines such as Ontop[15], RMLMapper[16], RocketRML[17], and MapSDI [18].

Research in semantic matching for automatic mapping generation can be characterized by two distinct eras: (i) the pre-LLM acceleration era and (ii) the post-LLM acceleration era. Before the rise of LLMs, semantic matching approaches primarily relied on textual and structural similarity techniques, employing string-similarity metrics (e.g., Levenshtein, Jaccard) and structural analyses (e.g., graph-based methods) to identify correspondences between schema elements [6, 19, 20]. However, these approaches often struggled to capture deeper semantic relations and contextual nuances. In the post-LLM acceleration era, LLMs have significantly reshaped the landscape through their capabilities in contextual representation, terminology disambiguation, and advanced semantic reasoning.

Hoseini et al. [5] investigates the challenges and opportunities of using LLMs to automate the creation of semantic models, particularly semantic type detection or automatic semantic labeling. Their approach identifies the meaning of table columns and links them to ontology concepts as labels. Using datasets from the VC-SLAM corpus and several LLMs (GPT-4, GPT-4o, and Llama-3), they show that adding contextual information to prompts and simplifying the ontology structure can improve the accuracy and performance of semantic type detection. Val-Calvo et al. [4] introduces OntoGenix, a tool designed to automate the construction of ontologies and Knowledge Graphs (KGs) from CSV datasets. OntoGenix provides modules for data processing, schema definition, ontology generation, and KG creation, all integrated into an LLM-based ontology engineering workflow. Nonetheless, OntoGenix struggles with complex scenarios and tends to produce a large number of annotations, leading to modeling pitfalls. Similar to [5], they also observe LLM hallucination issues. Moreover, the approach has not been fully validated on extensive datasets due to LLM token limitations.

Hofer et al. [9] utilizes LLMs to build a pipeline for generating RDF Mapping Language (RML) files that can self-configure, which is the first study to explore the creation of RML mapping files using LLMs. Their pipeline uses two prompts: the first generates the mapping, and the second corrects syntactic errors (verified using rdflib) in the result of the first prompt. Using the movie ontology and IMDB datasets (JSON), they experiment with models such as Claude 3 Opus, GPT-4, Claude 2.1, and GPT-3.5. Their results show that older LLMs frequently fail to produce valid RML documents and

**Table 1**
Comparison of related works and the research gaps addressed by LAMP.

| Paper | Year | Comparison & Research Gap |
| --- | --- | --- |
| Hoseini et al. [5] | 2024 | • Hoseini et al. [5] explores the potential and challenges of using LLMs, but does not validate generated outputs.
• LAMP provides a complete, practically usable mapping-file pipeline with LLM assistance.
• LAMP applies key insights from [5] (richer context and ontology reduction) and adds human validation for quality assurance. |
| Val-Calvo et al. [4] | 2024 | • OntoGenix focuses primarily on building new ontologies, even though it includes a mapping/KG population component.
• OntoGenix workflow is optimized for ontology creation rather than mapping to an existing ontology.
• LAMP is designed specifically for generating mappings from existing data sources to an existing ontology.
• LAMP uses retrieval-augmented function-calling for constrained outputs and breaks down the mapping workflow to improve accuracy and control. |
| Hofer et al. [9] | 2025 | • Hofer et al. and LAMP both generate mapping files, but Hofer et al. aims for full automation while LAMP maintains human oversight to address LLM limitations.
• Hofer et al. performs the mapping in a single prompt, whereas LAMP separates class mapping and property mapping. The decomposition reduces complexity when handling large ontologies and datasets.
• LAMP adds function-calling constraints and evaluation checkpoints to improve mapping correctness. |
| Schmidt et al. [3] | 2025 | • MYAM and LAMP both target mapping-file generation and use ontology reduction and retrieval to manage token limits.
• MYAM focuses on selecting a strong vector database for better retrieval, while LAMP prioritizes embedding models with richer vocabularies to improve similarity matching.
• MYAM enriches context with dataset-specific few-shot examples, whereas LAMP improves context through cleaned schema details and by embedding both class and property descriptions.
• LAMP provides global context at each stage, enabling the LLM to develop a coherent understanding of the class- and property-level mapping processes. |
| Castedo et al. [10] | 2025 | • BLINKG supplies datasets and evaluates various LLMs, but performs mapping with a single prompt.
• LAMP splits the mapping task into two stages and achieves better performance in both class and property mapping. |

that LLMs still struggle to map domain-specific properties consistently. Schmidt et al. [3] propose MYAM (Manufacturing YARRRML Mapping), an approach for generating YARRRML mapping files for manufacturing Knowledge Graph construction with the help of LLMs. Their method uses context-enhanced prompting (RAG-based retrieval of few-shot examples tailored to the dataset) and ontology reduction (via naïve and similarity-based methods). The evaluation shows that naïve ontology reduction combined with few-shot prompting yields the best results, where few-shot examples significantly improve precision. However, token limitations hinder scalability to large datasets, LLMs tend to be verbose, and commercial LLMs still struggle to interpret highly domain-specific terminology and abbreviations in complex domains, such as manufacturing.

Recently, Castedo et al. [10] introduced a dataset (BLINKG) for evaluating LLM performance in the mapping process, along with a gold-standard mapping serving as the baseline. Although BLINKG does not provide an in-depth discussion of the LLM evaluation results, the accompanying GitHub repository

includes performance results for several LLMs on the mapping task. Their approach relies on a single prompt to instruct the LLM to generate a complete mapping file. The generated file is then evaluated by comparing it against the predefined gold-standard mapping.

## 3. Methodology

In this section, we provide motivation for the LAMP design, an overview of the proposed pipeline, a detailed description of each component in LAMP, and an outline of the system design.

### 3.1. Design Motivation

At the beginning of our experiments, we followed the same strategy as [10] and [5], using a single prompt to ask the LLM to generate the entire mapping file. We explored multiple prompt variants and several prompting techniques, including structured prompts and few-shot examples. Although these improved the output, the results remained below our expectations, showing inconsistencies and declining performance as the ontology and data sources became more complex.

These limitations motivated us to decompose the mapping workflow into smaller, more manageable tasks: class mapping, property mapping, and final mapping-file generation. In LAMP, the smallest task given to the LLM is matching one concept from the data source to a small set of candidate ontology concepts. This design is the core principle of LAMP and is key to improving consistency and reliability in the mapping process.

Another advantage of task decomposition is its ability to overcome token limitations and handle large ontologies and data sources. However, this strategy also introduces at least three key challenges: (1) *how to determine the appropriate granularity for decomposition*—whether the model should map columns individually or process all columns of a table at once; (2) *how to provide sufficient contextual information for the LLM to perform each task effectively*; and (3) *how to ensure that outputs from different tasks remain coherent and interconnected.*

**Determine the appropriate granularity for decomposition.** To address the first challenge, we adopted several design decisions: First, we structured the mapping workflow into two levels: at the first level, tables are mapped to ontology classes; at the second level, each table's columns are mapped to the properties of the selected class. For Level-1 mapping, we employed a retrieval-augmented strategy using a vector database. This approach reduces the LLM's search space by requiring it to choose only from a small subset of the most semantically relevant classes. Specifically, we query the vector database using table-level information (the table name and selected metadata such as column names). The database is indexed such that each chunk contains one ontology class along with its property list and textual description. The retrieved candidate classes are then provided to the LLM, which is prompted to select the most conceptually appropriate class for the given table. This process is repeated for each table in the data source. Once table-to-class mappings are identified, we proceed to Level-2 mapping. At this stage, the LLM is prompted to map one or more columns of the table to the properties of the chosen class. This property-level mapping is performed per table, allowing the model to consider the full local schema context while maintaining manageable input size.

**Provide sufficient contextual information for the LLM.** To address the second challenge—providing sufficient but not excessive context—we observed that context size strongly influences both retrieval quality and prompting performance. Excessive context often reduces similarity scores during retrieval and may even confuse the LLM during prompting. Conversely, insufficient context leads to poor retrieval results, causing the model to miss the relevant candidate classes and making the downstream mapping task more difficult. To balance this trade-off, we applied the following strategies:
  1. **Term refining for table and column names**. Table and column names are often abbreviated and understandable only to domain experts. We therefore asked the LLM to expand these abbreviations

and generate short descriptions for each table using the table name, column names, and a few sample rows of raw data.

2. **Enriched representations for both queries and ontology chunks**. Each ontology class stored in the vector database includes not only its class name and textual description, but also its data properties, object properties, and IRI. Similarly, each query term (table) includes not only its original table and column names but also the refined terms generated in the previous step.

3. **Structured prompts for class mapping**. The class-mapping prompt follows a structured format consisting of: (i) a role definition, (ii) a task goal, (iii) explicit instructions, (iv) a global context (explained later), (v) the input term, and (vi) the candidate concepts retrieved from the vector database. Few-shot examples are also included to guide the model.

4. **Structured prompts for property mapping**. Property-level prompts are similarly structured and include: the role, goal, instructions, global context, the table name, the complete list of columns (including refined versions), the suggested set of relevant columns from class mapping, the selected ontology class, and its associated data and object properties. Few-shot examples are applied here as well.

**Ensuring that outputs from different tasks remain coherent and interconnected.** To address the third challenge—maintaining coherence across decomposed tasks—we introduced a global context that is included in every prompt, both for class mapping and property mapping. The global context contains a compact representation of the entire mapping space, including all table names, column names, ontology classes, and their properties. This mechanism ensures that the LLM retains a holistic view of the overall mapping process, even though tasks are executed independently. By providing this compact but comprehensive information, the LLM can better anticipate how a mapping decision relates to other decisions made at the same or different levels, thereby improving consistency and reducing contradictory outputs across tasks.

## 3.2. Proposed Pipeline Overview

The core mechanism of LAMP is the gradual use of LLMs to support semantic alignment between the data source and the target ontology. LAMP first performs higher-level mappings, where each table in the data source is mapped to one or more ontology classes. After these table–class correspondences are established, LAMP proceeds to finer-grained mappings by associating individual columns in each table with the corresponding ontology properties of the selected classes. Throughout both stages, the pipeline ensures that the LLM maintains awareness of the global context.

Although these mapping tasks are executed separately and in sequence, they are not treated as isolated subtasks. At each stage, LAMP supplies the LLM with a consistent mapping context, so that local decisions remain aligned with the overall semantic structure.

Figure 1 provides an overview of the LAMP. The LAMP comprises five stages in the mapping process: input, pre-processing, mapping definition, evaluation, and output. The processes within LAMP can be grouped into four categories: LLM-assisted tasks, embedding-based tasks, programmatically executed tasks, and human-assisted tasks. A human-in-the-loop component is retained to evaluate the LLM's outputs, ensuring the quality and reliability of the final results. The following subsection describes each core stage of the pipeline in detail.

### 3.2.1. LAMP Pre-Processing Stage

The pre-processing stage aims to prepare the ontology and the data source so that the LLM can interpret them more effectively during the mapping tasks. This stage consists of four main processes: extracting and transforming the ontology and data source; refining the terms (table and column names) from the data source; summarizing the schema to construct the global context; and chunking and embedding the concepts and terms. In this paper, the term *concept* refers to ontology classes, as ontologies

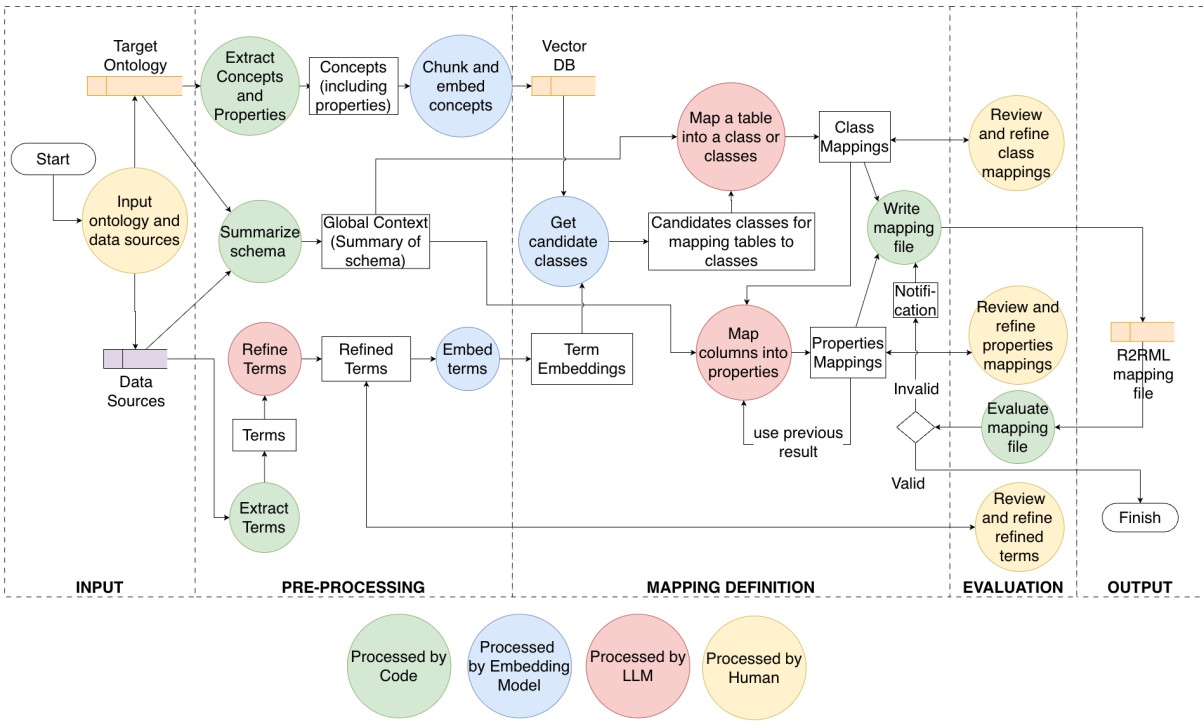

**Figure 1:** LLM-Assisted Mapping Pipeline Overview. *The diagram uses color-coded circles to indicate the category of each process: green circles represent processes executed programmatically; blue circles denote processes supported by an embedding model; red circles indicate processes assisted by an LLM; and yellow circles correspond to processes that involve human input.*

fundamentally conceptualize objects or entities. Meanwhile, the term *terms* refers to table and column names in the data source. The following subsections describe each of these processes in detail.

**Extracting and transforming.** In the target ontology, extraction focuses on classes and their related metadata, including labels, descriptions, definitions, full URIs, prefixed URIs, similar classes, data properties, and object properties. The extracted information is organized in a Markdown key–value (KV) format. Based on experiment [21], for tabular data, the Markdown KV representation is more readily interpretable by LLMs than alternative formats such as XML, JSON, or CSV. An example of a class represented in Markdown KV is shown in Figure 2a.

For the data source, extraction is performed at the table level. The extracted information includes the table name, column names, data types, primary-key and foreign-key information (i.e., related tables), and the first ten rows of data as samples. Unlike the target ontology, the transformation of the data source into Markdown KV format occurs only after the term-refinement step is complete. An example of a refined table represented in Markdown KV is shown in Figure 2b.

**Refining the terms.** This process enriches the semantic clarity of tables and columns by supplying additional contextual information that enables the LLM to understand their meaning better. The LLM is prompted to:

1. Expand any abbreviations or acronyms into their full, meaningful forms;
2. Propose clearer and more descriptive names for both tables and columns that more accurately reflect their intended semantics;
3. Translate these names into natural, grammatically well-formed English suitable for metadata documentation; and
4. Identify foreign-key relationships by analyzing each column for potential links to other tables.

As illustrated in Figure 2b, this step results in improved table names, enriched table descriptions, refined

```
# Route Class
**IRI**: http://vocab.gtfs.org/terms#Route
**Description**: A gtfs:Route is a commercial route followed entirely or partly by gtfs:Trips
**Prefixed IRI**: Route
## Data Properties
- **ID**: Type = string (IRI: http://www.w3.org/2001/XMLSchema#string)
- **URL**: Type = page (IRI: http://xmlns.com/foaf/0.1/page)
- **Long name**: Type = n3-114 (IRI: n3-114)
- **Short name**: Type = n3-117 (IRI: n3-117)
- **RouteDesc**: Type = string (IRI: http://www.w3.org/2001/XMLSchema#string)
- **Color**: Type = string (IRI: http://www.w3.org/2001/XMLSchema#string)
- **Text color**: Type = string (IRI: http://www.w3.org/2001/XMLSchema#string)
- **Sort Order**: Type = nonNegativeInteger (IRI: http://www.w3.org/2001/
XMLSchema#nonNegativeInteger)
## Object Properties
- **agency**: Domain = Route (IRI: http://vocab.gtfs.org/terms#Route), Range = Agency (IRI:
http://vocab.gtfs.org/terms#Agency)
- **Continuous Drop Off**: Domain = Route (IRI: http://vocab.gtfs.org/terms#Route), Range =
Concept, n3-7 (IRI: http://www.w3.org/2004/02/skos/core#Concept, n3-7)
- **Continuous Pickup**: Domain = Route (IRI: http://vocab.gtfs.org/terms#Route), Range =
Concept, n3-11 (IRI: http://www.w3.org/2004/02/skos/core#Concept, n3-11)
- **Route**: Domain = Trip (IRI: http://vocab.gtfs.org/terms#Trip), Range = Route (IRI:
http://vocab.gtfs.org/terms#Route)
- **Route type**: Domain = Route (IRI: http://vocab.gtfs.org/terms#Route), Range = Concept,
n3-28 (IRI: http://www.w3.org/2004/02/skos/core#Concept, n3-28)
```

```
# Table: Public Transport Routes\n
**Original Name**: routes\n
**Description**: This table represents public transport routes\n
**Column Count**: 9\n
## Columns\n
- **Route Identifier** (original: route_id)
- Type: VARCHAR(50)
- Examples: 4__1___, 4__10___, 4__11___ (identifier pattern)\n
- **Agency Identifier** (original: agency_id)
- Type: VARCHAR(50)
- Examples: CRTM, CRTM, CRTM\n
- **Route Short Name** (original: route_short_name)
- Type: TINYINT
- Examples: 1, 10, 11 (numeric pattern)\n
- **Route Long Name** (original: route_long_name)
- Type: VARCHAR(50)
- Examples: Pinar de Chamartín-Valdecarros, Hospital del Norte-Puerta del Sur, Plaza
Elíptica-La Fortuna (identifier pattern)\n
- **Route Description** (original: route_desc)
- Type: VARCHAR(255)\n
- **Route Type** (original: route_type)
- Type: TINYINT
- Examples: 1, 1, 1 (numeric pattern)\n
- **Route URL** (original: route_url)
- Type: VARCHAR(255)
- Examples: http://www.crtm.es/tu-transporte-publico/metro/lineas/4_1___.aspx,
http://www.crtm.es/tu-transporte-publico/metro/lineas/4__10___.aspx, http://www.crtm.es/
tu-transporte-publico/metro/lineas/4__11___.aspx (identifier pattern)\n
- **Route Color** (original: route_color)
- Type: VARCHAR(50)
- Examples: 2DBEF0, 005AA9, 009B3A\n
- **Route Text Color** (original: route_text_color)
- Type: VARCHAR(50) - Examples: FFFFFF, FFFFFF, FFFFFF\n
## Relationships\n
- **Agency Identifier** references **agency** table\n
## Data Characteristics\n
**Sample Size**: 10 rows\n
**Uniqueness**: route_id (all unique), agency_id (constant), route_short_name (all unique),
route_long_name (all unique), route_type (constant), route_url (all unique), route_color (all
unique)\n
**Domain**: transportation service',
```

|  a  |  b  |
| --- | --- |

**Figure 2:** Example of Markdown KV representation of (a) ontology and (b) terms from data source.

column names, identified relationships, and additional information describing the characteristics of each table.

**Chunking and embedding.**  For the ontology, this process reduces its adequate size by segmenting it into class-level chunks—based on the results of the ontology extraction and transformation step—and converting each chunk into vector embeddings, which are then stored in a vector database. For the data source, the same embedding procedure is applied to the refined terms for each table (produced by the extraction and term-refinement steps). These embeddings are later used to perform similarity-based queries against the vector database during the mapping process.

**Summarizing the schema.**  This process constructs the global context required for guiding the LLM throughout the overall mapping workflow. The global context provides a holistic view of the mapping process and serves as a connective layer between different mapping stages—both class-level and property-level tasks. Because a single prompt can hold only a limited number of tokens, the global context must be concise yet comprehensive. To generate this context, the schema summary extracts only class and property names from the ontology, whereas the data source includes only table and column names.

### 3.2.2. LAMP Mapping Definition Stage

In the mapping definition stage, four main processes are performed: obtaining candidate classes, mapping a table to one or more classes, mapping columns to properties, and writing a mapping file. The first process—**obtaining candidate classes**—aims to identify the most relevant ontology classes that will serve as possible mapping targets for each table. This step relies on a retrieval mechanism over the vector database, where each table term (augmented with additional contextual information) is converted into an embedding through the earlier chunking and embedding process, and then used as an embedded query. The embedded query is then used to retrieve semantically similar ontology concepts. This process is executed iteratively for each term, for every table in the data sources. The output of this step is a list of candidate ontology classes for each table, ranked by cosine similarity.

The second process—**mapping a table to one or more classes**—involves aligning each table from the data source with the most semantically appropriate ontology class or classes. This task is performed by prompting the LLM with a structured prompt that includes: (i) the global context summarizing the

overall schema; (ii) the detailed representation of the table being mapped; and (iii) the list of candidate ontology classes retrieved in the previous step. The LLM is then instructed to analyze the semantic relationships and determine the class or classes that best match the table, along with an explanation justifying its selection. When no suitable class exists, the LLM may explicitly state that no appropriate match is found. This process is repeated for each term, for every table in the data sources. The output of this step is a mapping between each table and its corresponding ontology class or classes.

After all tables have been mapped to ontology classes, the next step is **mapping columns to properties**, which focuses on establishing correspondences between individual columns in each table and the relevant ontology properties. Similar to the table-to-class mapping, this step is performed through an LLM prompt with the following structured input: (i) the global context; (ii) the detailed representation of the mapped table; (iii) the selected class or classes resulting from the table-to-class mapping; (iv) the complete list of columns and any suggested column–class associations; and (v) all data properties and object properties associated with the selected class.

Using this information, the LLM is instructed to analyze the semantic relationships and determine whether each column should be mapped to one or more ontology properties—either data properties or object properties. The LLM must also provide a justification for each mapping decision. As in the previous step, the LLM is allowed to refrain from mapping when no meaningful semantic alignment exists between a column and any available property. This process is applied independently to each table that has been successfully mapped in the preceding stage. The output of this step is a mapping between each column and its corresponding ontology property or properties.

Based on these two mapping outputs, the next step is to generate the **mapping file** that formalizes the mappings in a machine-readable format. The generated file is then validated through a syntactic verification process to ensure correctness and compliance with the expected mapping specification. Once validated, the mapping file can be used to populate the knowledge graph with data sourced from the underlying databases.

### 3.2.3. LAMP Evaluation Stage

In this study, evaluation is not positioned as a separate, post-processing stage. Instead, it is interwoven into specific points of the overall workflow. The evaluation is conducted both manually and programmatically, serving as a quality assurance mechanism to ensure the final mappings are reliable and ready for downstream use.

There are three human-involved evaluation steps: assessing the refined terms produced during the term refinement stage, validating the class-level mappings, and validating the property-level mappings. In each step, LAMP presents the LLM-generated output to the user, who may either revise the results or approve them directly. Figure 3 illustrates the LAMP interface, which captures user feedback during these evaluation stages.

Programmatic evaluation is applied during the generation of the mapping file. The generated mapping file is checked for syntactic correctness using the corresponding mapping-language library. Within LAMP, error messages returned by the library are leveraged to improve the system, reducing the likelihood of future errors during mapping file generation.

### 3.3. Tools and System Design

We implemented LAMP[1] using the following tools and configurations: (1) We decomposed LAMP into modular components—aligned with its internal processes—across the backend and frontend. The backend is responsible for the coding-intensive stages of LAMP, including interactions with the embedding model and LLM APIs. The frontend provides a user-facing interface for submitting inputs, visualizing progress, and offering feedback during the evaluation stages. (2) The backend is built using Python 3.9 with Flask. We also employ LangChain to simplify switching between embedding and LLM models, streamline prompt management, and, most importantly, ensure that the LLM's output follows

---

[1]The complete source code is available at the project repository: https://github.com/sulthoni/lamp

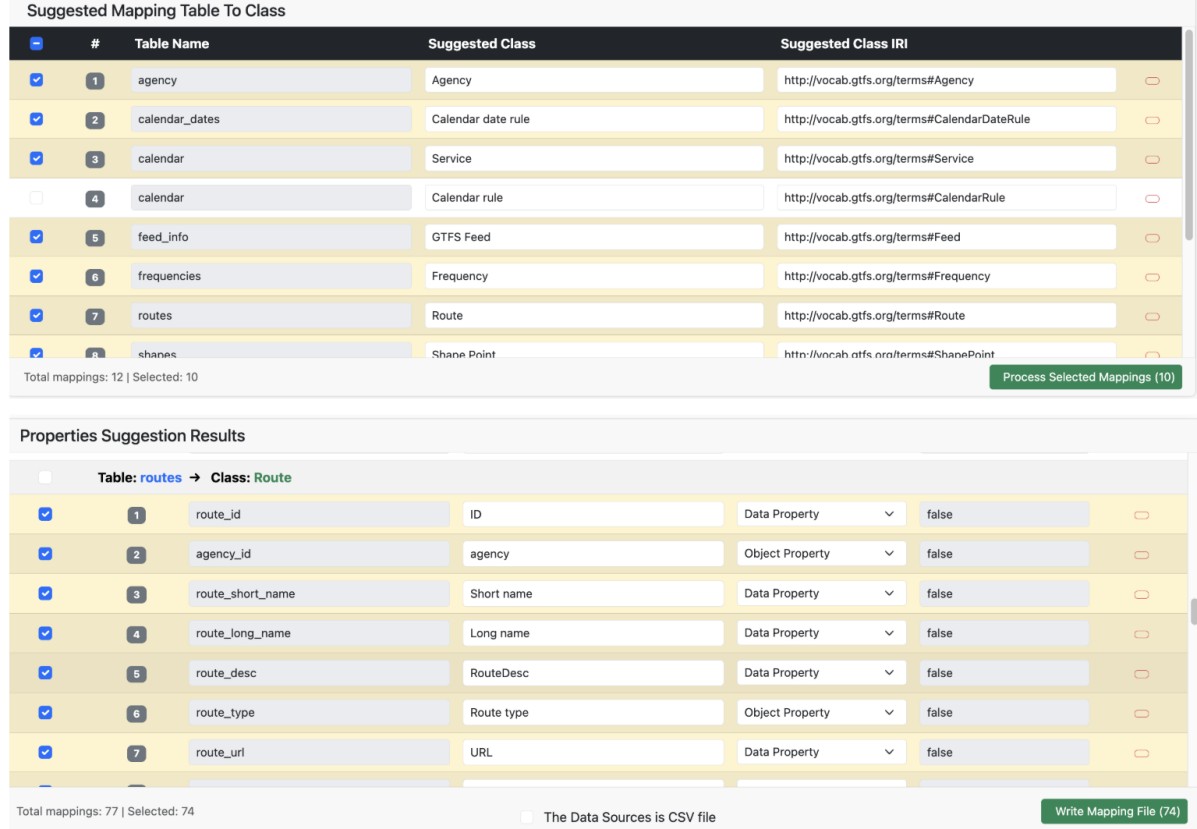

**Figure 3:** LAMP Interface to capture user feedback during evaluation stages.

the required format through function-calling capabilities supported by both the model and LangChain. (3) The frontend is developed using Angular 19, complemented by supporting libraries such as n3 for handling Turtle formats. Several extraction and transformation tasks are also performed on the frontend to reduce backend workload. (4) Through LangChain, LAMP provides multiple configuration options for LLMs, embedding models, and vector databases. However, the core requirement is that the selected LLM must support function calling to guarantee that the responses conform to the expected format. For this reason, we selected LLM models that support function calling and are comparable to those evaluated in BLINKG. We also use ChromaDB as the vector database. This selection was based on the model's support for function calling, its relatively large token limit and free API access, the broad domain coverage and high token capacity of gemini-embedding-001 (2,048 tokens), and the ease of use offered by ChromaDB.

## 4. Evaluation

We used the BLINKG dataset to evaluate the LAMP. BLINKG provides an ontology, source datasets, and a gold standard explicitly designed for benchmarking the ability of LLM-based systems to construct knowledge graphs [10]. The evaluation consists of three scenarios, each representing different levels of semantic and structural complexity:

- **Scenario 1 — Schema-Aligned Mapping**. This scenario examines LAMP's performance under conditions where the source data closely matches the target ontology. The dataset is characterized by close conceptual alignment and straightforward identification of ontology elements (classes, properties, and entities). As a result, the overall task complexity is low.

- **Scenario 2 — Functional and Partially Aligned Mapping**. In this scenario, the focus is on

**Table 2**
Characteristics, ontology name, number of classes, data properties, object properties in the gold standard, and number of tables and columns in the data source for each mapping scenario from the BLINKG dataset.

| Scenario | Characteristic | Ontology | Gold Standard | | | Data Source | |
| | | | Classes | Data Props | Obj Props | Tables | Columns |
|---|---|---|---|---|---|---|---|
| 1A | | | 1 | 1 | 0 | 1 | 1 |
| 1B | | | 1 | 2 | 0 | 1 | 2 |
| 1C | | | 2 | 2 | 1 | 1 | 2 |
| 1D | Low complexity, | Person, Sport, and | 1 | 4 | 0 | 1 | 3 |
| 1E | close match | City | 2 | 4 | 1 | 2 | 5 |
| 1F | | | 2 | 6 | 1 | 3 | 7 |
| 1G | | | 2 | 5 | 1 | 2 | 6 |
| 1H | | | 1 | 3 | 0 | 2 | 4 |
| 1 (total) | | | 12 | 27 | 4 | 13 | 30 |
| 2 | Medium complexity, value normalization | General Transit Feed Specification | 11 | 43 | 26 | 10 | 75 |
| 3 | High abstraction, minimal overlap | eProcurement Ontology (ePO) | 4 | 3 | 7 | 9 | 74 |

evaluating LAMP's ability to handle cases with moderate divergence between the data source and ontology. The dataset requires functional transformations—such as interpreting value formats, deriving meaning, or normalizing values—to achieve correct mappings. This places the scenario in the medium complexity category.

- **Scenario 3 — Schema-Distant and High-Abstraction Mapping**. The third scenario is the most challenging. The dataset exhibits minimal structural overlap with the target ontology and requires substantial contextual inference and semantic abstraction to align the data correctly. Due to these characteristics, this scenario demands strong abstract reasoning capabilities and reflects a high-complexity mapping environment.

Table 2 summarizes the characteristics of each mapping scenario, including the ontology name; the number of classes, data properties, and object properties in the gold standard; as well as the number of tables and columns in the data source. Scenario 1 comprises eight types, labeled 1A-1H, with increasing complexity at higher levels.

Several preparatory steps were carried out before using the datasets to evaluate LAMP: (1) For all scenarios in which the ontology input consisted of multiple files, we merged the ontology files by appending the additional ontology files to the end of the primary ontology file. (2) For Scenario 3, the input ontology did not comply with standard ontology principles—for example, several properties lacked domain and range definitions. We therefore corrected these issues by adding the missing domain and range information. (3) For Scenarios 1 and 3, the datasets were originally provided in XML and JSON formats. Before using them in the evaluation, we converted the data to a tabular format (CSV). In Scenario 1, this conversion was straightforward because each JSON file generally corresponded to a single CSV file. However, in Scenario 3, the dataset consisted of a single XML file, which we transformed into a tabular structure, resulting in nine CSV files.

## 4.1. Evaluation Metrics

To evaluate the mapping results produced by LAMP, we employed precision, recall, and F1-score metrics. These metrics are widely used in information retrieval and classification tasks to assess the accuracy and completeness of predicted results against a ground truth. Precision measures the proportion of correctly predicted mappings (true positives) out of all mappings predicted by the system. In contrast, recall

**Table 3**
Comparison of precision, recall, and F1-score across the mapping scenarios for class mapping, data property mapping, and object property mapping.

| Component | Metric | Scenario | | |
|---|---|---|---|---|
| | | 1 | 2 | 3 |
| Class | Precise | 1.0000 | 1.0000 | 0.6667 |
| | Recall | 1.0000 | 1.0000 | 1.0000 |
| | F1-Score | 1.0000 | 1.0000 | 0.8000 |
| Properties (Data + Object) | Precise | 1.0000 | 1.0000 | 0.3684 |
| | Recall | 0.9375 | 0.9855 | 0.7000 |
| | F1-Score | 0.9677 | 0.9927 | 0.4828 |
| Data Properties | Precise | 1.0000 | 1.0000 | 0.2727 |
| | Recall | 0.9286 | 1.0000 | 1.0000 |
| | F1-Score | 0.9630 | 1.0000 | 0.4286 |
| Object Properties | Precise | 1.0000 | 1.0000 | 0.5000 |
| | Recall | 1.0000 | 0.9615 | 0.5714 |
| | F1-Score | 1.0000 | 0.9804 | 0.5333 |

measures the proportion of correctly predicted mappings among all actual mappings in the ground truth. The F1-score is the harmonic mean of precision and recall, providing a single metric that balances both aspects. In this evaluation, we used Gemini 2.5 Flash as the LLM and gemini-embedding-001 as the embedding model.

## 4.2. Results

Table 3 summarizes the evaluation results of LAMP for both class mapping and property mapping (with data properties and object properties reported separately). The results are compared against the gold standard provided by BLINKG. In this evaluation, we assess only the semantic alignment produced by the LLM at the class and property mapping levels, without incorporating any user feedback or review. This setup allows us to observe the extent to which the LLM contributes to the mapping process within LAMP.

Table 3 shows that the LLM performs well in class mapping. In Scenarios 1 and 2, the F1-score reaches a perfect value. Only Scenario 3 yields an F1-score of 0.8, which is primarily due to additional class mapping suggestions, as indicated by the precision score of 0.67. These results demonstrate that LAMP is highly effective in handling class mapping.

For property mapping (combining both data and object properties), the results show that the LLM begins to struggle as scenario complexity increases. Scenarios 1 and 2 achieve F1-scores of 0.97 and 0.99, respectively, whereas Scenario 3 attains a substantially lower F1-score of 0.48. A more detailed analysis reveals that, across all scenarios, mapping data properties poses a greater challenge than mapping object properties. In Scenario 1, the F1-score for data property mapping is 0.96, compared to 1.0 for object property mapping. This discrepancy arises because the LLM occasionally misses certain column-to-data-property mappings, leading to a lower recall score of 0.93. However, the model consistently achieves perfect precision in this scenario.

In Scenario 2, which represents a medium level of difficulty, the LLM continues to perform very well, achieving an overall F1-score of 0.99. A closer inspection shows that this value is influenced by the recall score for object property mapping, which is 0.96. This indicates that the LLM missed a small number of object properties mapping in Scenario 2.

In Scenario 3, the LLM's performance in mapping both data and object properties is moderate, with F1-scores of 0.43 and 0.53, respectively, yielding an overall property-mapping F1-score of 0.48. A deeper analysis reveals that, in addition to missing some column-to-property mappings, the lower F1-scores are also due to the LLM producing more property mapping suggestions than those in the gold standard. The raw mapping outputs show that the LLM frequently proposes mappings from identifier-related

**Table 4**
Comparison of F1-scores (Averaged per Scenario)

| Model | BLINKG - Avg. F1 (Class & Properties) | | | LAMP - Avg. F1 (Class & Properties) | | |
|---|---|---|---|---|---|---|
| | Scenario 1 | Scenario 2 | Scenario 3 | Scenario 1 | Scenario 2 | Scenario 3 |
| Deepseek R1 | 0.9881 | 0.8623 | 0.4537 | 0.9146 | 0.8646 | 0.4375 |
| Gemini 2.5 Pro | 1.0000 | 0.9792 | 0.3479 | 1.0000 | 1.0000 | 0.7077 |
| GPT-4o | 0.9931 | 0.9781 | 0.1967 | 0.9364 | 0.9454 | 0.4500 |
| Llama 3.3 70B | 0.9795 | 0.9139 | 0.2713 | 0.9259 | 0.8980 | 0.5105 |
| Mixtral 8x22B | 0.9801 | 0.9000 | 0.1064 | 0.9655 | 0.9355 | 0.4156 |
| OpenAi o3 | 0.8546 | 0.9653 | 0.4948 | 0.9561 | 0.9652 | 0.9652 |

**Table 5**
Ablation Study Result

| LAMP Features | F1 - class mapping | | | F1 - property mapping | | |
|---|---|---|---|---|---|---|
| | 1G | 2 | 3 | 1G | 2 | 3 |
| Full features | 1.00 | 1.00 | 0.80 | 1.00 | 0.99 | 0.48 |
| Without Global Context | 1.00 | 0.95 | 0.67 | 0.91 | 0.98 | 0.40 |
| Without Term Refinement | 1.00 | 0.95 | 0.75 | 0.91 | 0.96 | 0.45 |
| Without Enriched Representation | 1.00 | 1.00 | 0.67 | 0.91 | 0.98 | 0.54 |
| Without Process Decomposition | 1.00 | 1.00 | 0.33 | 0.80 | 0.99 | 0.13 |

columns to data properties (e.g., `epo:hasIdentifierValue`), even though such mappings are not present in the gold standard.

### 4.3. Comparison of LAMP and BLINKG mapping results

In BLINKG, the mapping process was evaluated using several LLM models—Deepseek, Gemini 2.5 Pro, GPT-4 Omni, Llama-3.3-70B, Mixtral 8x22B, and OpenAI o3 [10]. Unlike LAMP, BLINKG did not decompose the mapping workflow into multiple stages; instead, it performed mapping with a single prompt to the LLM. To enable a fair comparison with BLINKG, the second evaluation uses the same LLM models as those reported in BLINKG. However, since LAMP additionally incorporates an embedding model, we evaluate combinations of LLM and embedding models as follows: (1) DeepSeek R1 with qwen3-embedding-8b, (2) Gemini 2.5 Pro with gemini-embedding-001, (3) GPT-4 Omni with text-embedding-3-large, (4) Llama-3.3-70B with pplx-embed-v1-4b, (5) Mixtral 8x22B with llama-nemotron-embed-vl-1b-v2, and (6) OpenAI o3 with text-embedding-3-large. Table 4 presents the comparison of the average F1-scores (for both class and property mapping) per scenario between BLINKG and LAMP.

The comparative results demonstrate that LAMP outperforms BLINKG across most LLM models, particularly in Scenario 3. In contrast, for Scenarios 1 and 2, the performance of both approaches is comparable, with only marginal differences–sometimes slightly higher, sometimes slightly lower. In Scenario 3, however, the improvement in F1-score is substantial for five models (excluding Deepseek R1), with the highest score of 0.9652 achieved by OpenAI o3. The results demonstrate that the decomposition strategy employed in LAMP effectively reduces the complexity of mapping tasks, particularly in more challenging scenarios.

### 4.4. The impact of each feature on the overall performance

To assess the contribution of each LAMP feature to the overall performance, we conducted an ablation study. This approach systematically removes a specific feature from LAMP to assess its impact on the resulting F1 Score. Table 5 presents the results of the ablation study on the BLINKG dataset using Gemini 2.5 Flash and gemini-embedding-001 across Scenarios 1G, 2, and 3.

Table 5 shows the result of our ablation study: the first row reports the F1-scores obtained when

LAMP is executed with all features enabled; the second row presents the results without the global context; the third without term refinement, the fourth without enriched representation; and the last row without process decomposition.

The results clearly indicate that removing process decomposition has the most significant negative impact. When LAMP performs mapping using a single prompt for both class and property mapping—without decomposing the workflow—the F1-score drops substantially, from 0.80 to 0.33 for class mapping and from 0.48 to 0.13 for property mapping. This demonstrates that decomposition is the most critical design component of LAMP.

The second most influential feature is the global context. When the global context is removed, the F1-score decreases from 0.80 to 0.67 for class mapping and from 0.48 to 0.40 for property mapping. These findings confirm that maintaining cross-task coherence via a global context representation significantly improves overall mapping performance.

## 5. Discussion

Selecting and employing a more capable and intelligent LLM is indeed a primary factor in achieving strong mapping performance, as consistently shown in prior studies [5, 9, 10]. However, with a well-designed pipeline—incorporating process decomposition, effective context management, and other structured components—can significantly improve the mapping performance of LLMs. Our evaluation of the LAMP approach demonstrated this hypothesis. By employing its structured approach, LAMP achieves notably better performance in complex scenarios compared to BLINKG, which relies on a single-prompt strategy, yielding more reliable and accurate results.

Task decomposition is the core mechanism behind LAMP and the primary source of its performance gains, aligning with evidence from [22] that structured decomposition substantially boosts accuracy in LLM-based systems. A second factor is the degree of contextual sufficiency: class mapping consistently outperforms property mapping because it provides the model with a richer and more comprehensive informational basis for reasoning.

**Human Involvements.**  As task complexity increases, the LLM's performance tends to diminish. Likewise, when the model lacks sufficient knowledge of domain-specific concepts, its ability to generate accurate mappings also decreases. Consequently, LAMP includes human review to validate and refine outputs as needed. At the same time, we recognize that LLMs continue to evolve rapidly and are becoming increasingly capable. It is therefore plausible that, in the future, the quality of LAMP's automatic mapping suggestions will improve to the point where human intervention can be further reduced. To this end, in either scenario—under current limitations or future advances—LAMP remains a viable and adaptive solution that can operate effectively across different stages of LLM maturity.

**Error Analysis.**  As discussed in the results section, two types of mapping errors were observed: (i) cases in which the LLM failed to produce mappings that align with the gold standard (reflected in lower Recall), and (ii) cases in which the LLM generated mapping suggestions that were not present in the gold standard (captured through Precision). During our initial experiments—prior to introducing the global context—we observed noticeably lower Recall and Precision. Several mapping outcomes revealed missing class mappings and duplicated property mappings. Missing class mappings typically occurred when a table contained substantial information related to two different classes, leading the LLM to map it only to the dominant one. Meanwhile, duplicated property mappings often appeared in datasets that included join tables linking two or more entities.

The introduction of the global context allowed the model to make more informed decisions during each individual step. As a result, the LLM could better infer which class was mapped in earlier tasks and whether a particular property mapping had already been assigned elsewhere. Further, the inclusion of the global context mitigated both types of errors and strengthened the consistency and accuracy of the overall mapping process.

Another observation from our experiments is that the low F1-score for property mapping in Scenario 3 is mainly attributable to the incompleteness of the input ontology. Therefore, ensuring that the ontology is well-structured, explicit, and semantically complete is crucial for achieving high-quality mappings, particularly in scenarios requiring property-level alignment.

**Limitations.** Although LAMP achieves higher F1 scores than existing methods, it still has several limitations. First, when LAMP decomposes the workflow into smaller tasks, the overall token usage (i.e., the cumulative tokens across all class-mapping and property-mapping steps) increases. For LLM providers that charge based on token consumption, this leads to higher operational costs. Second, LAMP currently depends on LLMs that support function calling; without this capability, reformatting the model's raw responses requires significantly more manual effort. At present, LAMP is also limited to tabular data sources or those that can be transformed into tabular form. Finally, the supported mapping language is restricted to R2RML, limiting its applicability in environments that rely on alternative mapping standards.

## 6. Conclusion and Future Work

This paper presents LAMP, a novel LLM-based approach for ontology-based data mapping. By decomposing the mapping workflow into structured subtasks—supported by term refinement, targeted retrieval, careful management of contextual sufficiency, and a shared global context—LAMP effectively enhances the accuracy of semantic mappings. Using the BLINKG dataset, we evaluated LAMP and observed clear improvements over existing single-prompt approaches, achieving higher F1-scores for both class and property mappings across varying levels of complexity.

In the future, LAMP can be extended with agentic capabilities and integrated memory mechanisms. Each task could be assigned to a specialized agent, while an agent manager would coordinate the overall pipeline. The evaluation process could also be orchestrated by the agent manager, who could introduce a dedicated evaluation agent to perform intermediate checks before the final human review. Incorporating memory functions would further reduce reliance on manually constructed global context, allowing LLM agents to dynamically determine how much contextual information to pass to subsequent agents. This would enable a more adaptive, scalable, and autonomous mapping workflow.

## Declaration on Generative AI

During the preparation of this work, the author(s) used GPT-5.1 for Indonesian–English translation, grammar and spelling checks, and for refining and clarifying explanations. After using these tool(s)/service(s), the author(s) reviewed and edited the content as needed and take(s) full responsibility for the content of the publication.

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
