# OpenReview forum: "Reducing Human Effort in Ontology-Based Data Integration: An LLM-Assisted Mapping Pipeline"
_eswc-conferences.org/ESWC/2026/Workshop/KGCW — KGCW 2026_

### Official Review · ~Herminio_García-González1 · 2026-03-20
**Interesting but it would benefit from some clarifications**

**Rating:** 7
**Confidence:** 4

**Review:**

This paper introduces a new tool for the semi-automatic creation of mapping rules using schema information (ontology and relational DB schemas) as inputs for an LLM-based workflow. The approach is well grounded on recent advancements and trends within the LLM ecosystem such as agentic task division and context management, and has been evaluated against a recently-released benchmark (BLINKG), used as ground truth. As such, the paper fits well under the scope of the workshop and should be of interest for its audience while helping to spark some discussions about the automation of KGC workflows.

Strengths

- Well aligned with the workshop topics and scope
- Well grounded in recent advancements and literature both in KGC and LLMs.
- Code available and evaluation based on public benchmark
- Useful for other practitioners

Limitations

- Only ontologies and relational DB schemas are supported as inputs
- Evaluation not using the same models and slightly limited
- Uncertainty about its deployment in privacy-first and limited resources scenarios

While the paper scope and reach are in my opinion sufficient for this venue I leave here some observations and remarks that could help the authors in preparing the final version of this paper or considering this as future work:

My main concern comes with the context management as it seems to be one of the key features for improving the performance w.r.t. to the baseline method. Due to privacy issues some organisations are starting to rely on open source and locally deployed LLMs, however due to hardware restrictions many of these models face serious limitations, being one of those the context window size. Without elaborating further on this topic the reach of this tool is somewhat limited, leaving it to open source domains or big organisations that could ensure running big models in house.

The comparison with the baseline method (Table 3) seems a bit limited and slightly biased. Mainly, because LAMP was evaluated using Gemini 2.5 Flash and BLINKG against Gemini 2.5 PRO and GPT4o. Being the LLM at the heart of this development, evaluating over the same model would constitute a fairer comparison. At the same time, the reason not to include the rest of the methods in Table 1 should be highly emphasised.

As for the pipeline itself, I was wondering the effect that employing a GraphRAG approach would have for the retrieval-augmented strategy described in Section 3.1. Given that the ontology is essentially a graph, the authors could try to implement a more fine-grained lookup method. Similarly, looking at Figure 2, the formats do not seem to align very well, despite the LLM being able to reconcile them afterwards. Would the introduction of a previous phase for schema alignments improve the results? Regarding the general context and ignoring the context size limitations, what difference would entail for the process and final output having the whole history of operations (similar to the chat history) available for the LLM instead of providing the general context at each step?

Finally, some formatting suggestions:

- Related work: put the references next to the mentioned tool and not at the end of the sentence.
- Section 3, last bullet point: Use numbers as in the previous item.
- Figure 2: Use listings(?)
- Figure 4: Best to use a table for this data.
- Section 4.4, second paragraph: Hard to follow how each feature corresponds to the steps in Figure 1.
- LAMP acronym inevitably made me think of the other LAMP (Linux, Apache, MySQL, PHP) at first sight. You might want to consider a different acronym down the line.

---

### Official Review · ~Jakub_Klímek1 · 2026-03-20
**Addresses a current challange in a rather straight-forward way**

**Rating:** 5
**Confidence:** 4

**Review:**

In this paper, the authors present LAMP - an LLM-assisted pipeline for mapping data to ontologies, supporting the process of OBDI - Ontology-Based Data Integration. The main focus is on splitting a task into several stages (decomposition), while repeatedly providing an overall context in each LLM prompt. This is a standard optimisation technique whenever working with LLMs and larger documents/data, as evidenced in the related work section. The approach is then evaluated using the BLINKG benchmark.

The strengths of the paper include:
1. The topic fits well with the workshop
2. The approach demonstrates the benefits of decomposing LLM-based tasks

The weaknesses:
1. While in the introduction, the authors make it seem like LAMP is focused on heterogeneous data for the integration task, later, in 3.1, it becomes evident that only tabular data is supported, and the whole approach is fitted to tabular data (columns and rows phases). This is then proven in 4, where the authors somehow transform the BLINKG JSON and XML data into CSV, before running LAMP. This should be explicitly said, that LAMP is limited to tabular data.
2. The transformation of the benchmark data into CSV files is not described sufficiently, and the files are not provided in the GitHub repo, making the results irreproducible
3. In Evaluation, the LAMP approach is only compared to the BLINKG approach, which only uses one LLM prompt for the mapping task. IT then seems quite straightforward that the LAMP approach will perform better thanks to the decomposition and global context techniques. It would be more interesting to compare the LAMP approach to comparable related work, also applying decomposition.
4. The implementation could benefit from supporting not only cloud-based commercial LLMs, but also local LLMs, e.g., using Ollama

---

### Official Review · ~Gertjan_De_Mulder1 · 2026-04-02
**Practically relevant solution with room for improvements**

**Rating:** 6
**Confidence:** 3

**Review:**

The paper proposes an LLM-based mapping pipeline that decomposes the mapping task into smaller tasks while incorporating a global context to maintain coherence. The solution is evaluated using the BLINKG benchmark across three scenarios representing increasing levels of semantic and structural complexity.

The main contribution of the paper is an applied, decomposition-based LLM pipeline for ontology mapping.

**Strengths**

The paper is well-written and well-structured.

The paper addresses a real and practically relevant problem.

The evaluation’s ablation study provides meaningful insights on each component’s impact on performance.

**Weaknesses**

The current evaluation is confined to a single LLM (Gemini 2.5 Flash), which limits assessment of the generalized impact of the proposed pipeline.

The authors mention increasing token consumption due to task decomposition as a limitation. Given the practical setting of this paper, a preliminary evaluation of token consumption would be beneficial.

The paper’s title claims a reduction in human effort, however, this claim is not directly supported in the evaluation/results.

---

### Decision · Program_Chairs · 2026-04-09

**Decision:**

Accept

**Comment:**

This paper has been selected for presentation at the KGC workshop. We strongly encourage the authors to consider the reviews whilst revising the paper. Camera-ready instructions will soon follow.